# PHYSMCA: PHYSICAL MULTI-MODAL BACK-DOOR–ADVERSARIAL EXAMPLE COLLABORATIVE ATTACK

## ABSTRACT

The multi-modal visual system relies on deep learning models to enable all-weather perception, but it faces the dual security threats of backdoor and adversarial attacks. Although targeted defense methods have been proposed, these studies mostly focus on a single modality or a single scene, and there is still a lack of systematic exploration of the physical differences across modalities and the collaboration mechanism between backdoor and adversarial attacks, thereby exposing exploitable vulnerabilities. To this end, we propose physical multi-modal backdoor-adversarial example collaborative attack (**PhysMCA**) to achieve end-to-end design from digital domain optimization to physical domain verification. Our method includes two key innovations: (1) a multi-modal trigger localization algorithm for the chest-abdominal region based on human pose prior and Bayesian Optimization is proposed. Compared with traditional static template matching, the proposed method improves the accuracy, adaptability, and robustness of multi-modal target localization; (2) a joint optimization of backdoor implantation and adversarial perturbation in multi-modal models is proposed based on micro composite trigger and lightweight adversarial camouflage mechanism, which forms a multi-modal attack link with high concealment and poses significant challenges to existing single-attack detection mechanisms. The experimental results show that our method has excellent performance in both digital and physical domains. In the physical domain, the attack success rate in the visible light reaches 93.7%, and the attack success rate in the thermal infrared reaches 90.4%.

## 1 INTRODUCTION

With the rapid development of deep learning (DL) technology (Krizhevsky et al., 2017), the computer vision systems based on DL models have been widely used in key fields such as safety monitoring, intelligent transportation, and medical diagnosis (Azfar et al., 2024; Esteva et al., 2021). However, the inherent vulnerability of DL models (Qiu et al., 2019) has gradually become a serious latent threat limiting their reliable deployment. Such vulnerabilities can be exploited not only to achieve malicious objectives but also to expose sensitive information about critical personnel and key assets in the public domain, causing serious privacy risks (Zhang et al., 2023). Among various attack methods, the backdoor attack (Gu et al., 2019) and the adversarial attack (Szegedy et al., 2013) are two typical research paradigms that can compromise the decision integrity of models in a non-overlapping manner. The fusion of these two attack paradigms - known as the collaborative attack - can broaden the attack surface and enhance the robustness of attacks, and has become a frontier topic in the field of DL security (Liu et al., 2021).

Most current research on backdoor attacks and adversarial attacks focuses on single-modality data (Li et al., 2022). However, in practical scenarios such as night safety monitoring and thermal imaging detection, multi-modal systems are typically employed to improve environmental adaptability (Zhang et al., 2019). Thermal infrared (IR) images, which capture the thermal radiation of objects, are less affected by illumination and occlusion, whereas visible light (VL) images provide detailed texture information (Li et al., 2018) (for brevity, we refer to "thermal infrared" as "infrared" and "visible light" as "visible" throughout the paper). This complementarity makes multi-modal computer vision systems more reliable in complex environments (Zhang et al., 2020), but it

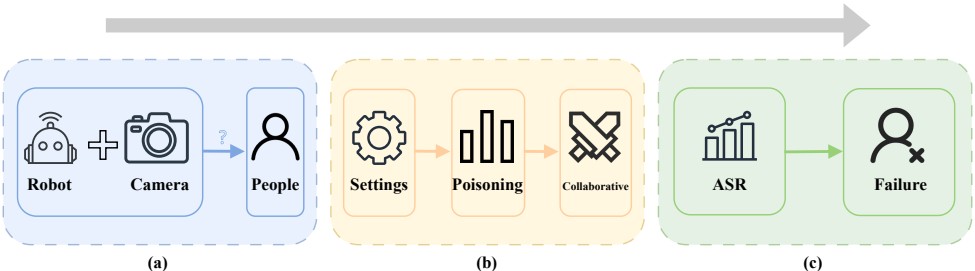

Figure 1: **Overview of PhysMCA attack process.** (a) Multi-modal pedestrian detection scenario, (b) collaborative attack framework, (c) pedestrian detection failed.

also brings new challenges to attack research: first, the discrepancies between infrared and visible modalities lead to poor transferability of attack strategies between the two modes (Zhu et al., 2021); secondly, most of the existing backdoor triggers are designed for the digital domain (Wenger et al., 2021), and their effectiveness in the physical domain has not been fully verified in multi-modal scenarios. Finally, it is still unclear what effect the collaborative mechanism of backdoor attack and adversarial attack in multi-modal systems will have.

In order to cope with the above challenges, we propose a multi-modal backdoor-adversarial collaborative attack framework in the physical domain for the first time, as shown in Figure 1. It focuses on three core issues: (1) How to optimize the trigger position in the digital domain; (2) How to determine the optimal poisoning rate of multi-modal backdoor training; (3) How to design a collaborative attack mechanism in the physical domain and evaluate the effectiveness of the backdoor-adversarial collaborative attack in both digital and physical domains (including infrared and visible modalities).

Our contributions are summarized as follows:

1. Trigger position optimization based on human pose estimation and Bayesian Optimization algorithm: First, MediaPipe pose detection is employed to restrict the trigger placement to the chest and abdomen regions of the human target. Subsequently, the Bayesian Optimization algorithm is employed to search for the optimal trigger position within the restricted region, enabling adaptive optimization of the trigger placement in both visible and infrared modalities.

2. A multi-modal poisoning rate optimization strategy: We construct a self-compiled multi-modal dataset and supplement it with public datasets. For single modality (infrared or visible) and mixed modality (infrared plus visible), the optimal poisoning rate is trained, evaluated and selected. On this basis, we further propose a multi-modal comprehensive loss function to estimate the optimal poisoning rate.

3. A systematic and comprehensive evaluation of collaborative attack in the digital and physical domains: The pedestrian detection model optimized according to the optimal poisoning rate is used as the main test platform. We construct a three-layer evaluation framework comprising single-attack verification, collaborative-attack verification, and robustness-enhancement verification. Finally, a multi-modal attack evaluation standard is established, covering the entire process from digital optimization to physical deployment.

## 2 RELATED WORK

**Backdoor attack:** Backdoor attack poisons machine learning models during training by embedding a small number of trigger-embedded samples in the training dataset. The model behaves normally with clean inputs but outputs attacker-specified results when a trigger is present. BadNets (Gu et al., 2019) pioneered digital domain backdoor attacks by modifying local image pixels and associating them with target labels. At present, research on digital domain backdoor attacks has reached a stage of systematic development: label pollution attacks (Gu et al., 2019; Turner et al., 2019; Saha et al., 2020); trigger signal-specific attacks (Li et al., 2021; Wang et al., 2024); attack performance optimization (Doan et al., 2021; Souri et al., 2022), and scene innovation focusing on classification tasks (Xue et al., 2022b; Yin et al., 2025) have all achieved fruitful results, reflecting the trend toward

diversified development in this field. Currently, researchers in the physical domain have conducted a series of targeted studies on trigger design (Yin et al., 2024a; Xue et al., 2022a) and conceal-ment (Han et al., 2022; Dao et al., 2024), achieving notable interim results. However, whether in the digital domain or physical domain, backdoor attacks are mostly limited in scope to the visible spectrum. Yin et al. (2024b) systematically analyzed and experimentally validated trigger designs in the physical domain. However, the trigger in this scheme requires additional power supply and heating, and has the defects of large volume and weight. In response to this problem, we proposes a new implementation of multi-modal backdoor attacks in the physical domain. Specifically, we use low-emissivity aluminum foil and high-precision color-printed cardboard to fabricate a badge-sized trigger, which effectively addresses the limitations of existing solutions regarding power supply, size, and weight.

**Adversarial attack:** Adversarial attack is a typical attack paradigm for machine learning models. Its core mechanism is to mislead the model into making incorrect decisions by constructing small and imperceptible input disturbances, thus destroying the reliability of the model and the security of practical applications. Szegedy et al. (2013) first proposed the concept of adversarial samples in 2013, which laid the foundation for research in this field. With the advancement of research, adversarial attacks form two core classification systems: in terms of attack objectives, it can be divided into targeted attack (Goodfellow et al., 2014; Zolfi et al., 2021; Zhou et al., 2023) and untargeted attack (Hu et al., 2023; Zhao et al., 2024; Cheng et al., 2024); in terms of the attacker's knowledge of the model, it can be divided into white box attack (Gao et al., 2024; Aggarwal et al., 2024) and black box attack (Zhou et al., 2025; Reza et al., 2023). Early research on adversarial attacks mostly focused on the visible scene in the digital domain (Szegedy et al., 2013; Goodfellow et al., 2014; Zolfi et al., 2021). In recent years, the research boundary has gradually expanded to the physical domain and infrared mode (Wei et al., 2023; Cheng et al., 2024) and has made phased progress. However, the current research on the collaboration mechanism of multi-modal backdoor and adversarial attacks in the physical domain is still in a blank state and lacks systematic research results.

## 3 METHOD

This section introduces our method in detail, as shown in Figure 2, the PhysMCA framework con-sists of a three-tier framework: multi-modal adaptive trigger localization, poisoning rate optimiza-tion, and cross-domain joint attack implementation. The framework provides an end-to-end attack design from data modeling to physical verification, with each module rigorously modeled and opti-mized using mathematical principles.

### 3.1 ADAPTIVE TRIGGER LOCALIZATION BASED ON HUMAN POSE PRIOR AND BAYESIAN OPTIMIZATION

There is a significant trade-off between the physical concealment of the trigger and the effectiveness of the attack: if the trigger is placed too conspicuously, it is easy to be observed; if the position devi-ates from the key regions detected by the model, the attack effect will be weakened. To resolve this trade-off, we propose an adaptive positioning framework that integrates human pose estimation and Bayesian Optimization. Candidate regions are first constrained by anatomical priors; subsequently, the optimal position is determined via a data-driven optimization strategy.

**Human detection and preliminary positioning of key areas:** First, the pedestrian detection model is employed to detect human targets in the image and extract the corresponding bounding boxes. Then, the MediaPipe Pose model (Bazarevsky et al., 2020) is used to extract 14 key points of the human body, as shown in Figure 3 (a). Additionally, in order to improve the detection stability of low-quality images, a multi-scale enhancement strategy is adopted: the input image is scaled, and the coordinates of the detection results of each scale are calibrated, finally, the key point set with the highest visibility is retained.

**Adaptive adjustment of trigger candidate region based on human pose prior:** Under infrared conditions, the most stable temperature regions of the human body are the chest and abdomen (Davis & Sharma, 2007), Therefore, we constrain the trigger to the torso area to balance the attack effec-tiveness and concealment. Using the estimated pose keypoints, we first compute the torso scale and

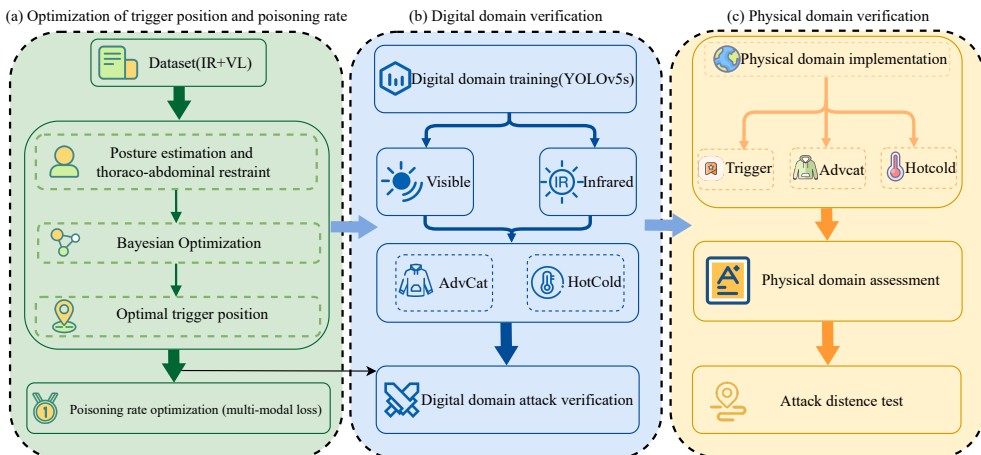

Figure 2: The PhysMCA overall framework.

then adapt the trigger size accordingly. To ensure physical consistency, the trigger height $h_t$ is set as a fixed proportion of the human bounding-box height.

**Optimal trigger location search based on Bayesian Optimization:** In the candidate region, Bayesian Optimization (BO) (Snoek et al., 2012) is used to find the trigger center coordinate $(c_x, c_y)$ that makes the attack effect optimal. BO models the uncertainty of the objective function through Gaussian process (GP), and efficiently balances the search for unknown regions and the use of known regions. Next, the search space is defined, and the trigger center needs to be limited to the chest and abdomen areas, that is:

$$X = [x_{\text{chest,min}}, x_{\text{chest,max}}] \times [y_{\text{chest,min}}, y_{\text{chest,max}}] \cup [x_{\text{abdomen,min}}, x_{\text{abdomen,max}}] \times [y_{\text{abdomen,min}}, y_{\text{abdomen,max}}], \quad (1)$$

Where $[x_{\text{chest,min}}, x_{\text{chest,max}}]$ and $[y_{\text{chest,min}}, y_{\text{chest,max}}]$ represent the horizontal and vertical boundaries of the chest region in the image, respectively. $[x_{\text{abdomen,min}}, x_{\text{abdomen,max}}]$ and $[y_{\text{abdomen,min}}, y_{\text{abdomen,max}}]$ correspond to the horizontal and vertical boundaries of the abdominal region in the image, respectively. Ensure that the trigger is completely located within the candidate region. Finally, Bayesian Optimization is performed: the objective function is fitted by GP, the next evaluation point is selected by collecting Expected Improvement (EI), and the GP modeling and acquisition function optimization steps are repeated to obtain the optimal position. The experiment combines the comprehensive results of infrared and visible, and selects the left chest position as the optimal trigger position, as shown in Figure 3 (b) and (c) (the detailed derivation process is shown in Appendix A).

### 3.2 MULTI-MODAL POISONING RATE OPTIMIZATION

The poisoning rate $\eta$ directly determines the balance between the model's backdoor activation ability and the clean sample performance. For the multi-modal scene of visible and infrared images, we first define a multi-modal loss function to quantify the trade-off between the attack success rate (ASR) and the clean sample confidence (C). Through analytical derivation and verification on a small number of discrete points, we approximately obtain the optimal $\eta^*$.

**Definition of loss function:** The multi-modal comprehensive loss function $\mathcal{L}(\eta)$ is constructed as follows :

$$\mathcal{L}(\eta) = \alpha \cdot [2 - \text{ASR}_{\text{IR}}(\eta) - \text{ASR}_{\text{VL}}(\eta)] + \beta \cdot [\max(0, \epsilon - C_{\text{IR}}(\eta)) + \max(0, \epsilon - C_{\text{VL}}(\eta))], \quad (2)$$

Where $\text{ASR}_{\text{IR}}(\eta)$ and $\text{ASR}_{\text{VL}}(\eta)$ are the backdoor ASR of the model on the infrared and visible modalities test sets, respectively; and $C_{\text{IR}}(\eta)$ and $C_{\text{VL}}(\eta)$ are the average detection confidence of the model on clean infrared and visible samples, respectively. $\alpha$ and $\beta$ are weight coefficients, and $\epsilon$ is the minimum clean sample confidence threshold. The first item penalizes the low ASR in both modals (the target is to make $\text{ASR}_{\text{IR}}(\eta)$ and $\text{ASR}_{\text{VL}}(\eta)$ approach 1, thereby making this item approach 0). If the clean sample confidence is lower than $\epsilon$, the second item imposes a penalty to ensure the normal performance of the model on clean data.

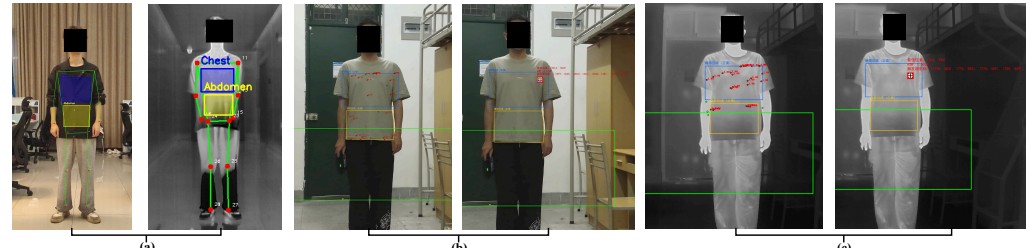

Figure 3: (a) Demonstration of human pose key point detection, (b) demonstration of trigger optimization results.

**Constrained optimization of poisoning rate:** Under the constraint that $\eta$ belongs to the effective set $\Omega$, by solving the optimal poisoning rate $\eta^*$:

$$\eta^* = \arg\min_{\eta \in \Omega} \mathcal{L}(\eta). \tag{3}$$

In order to verify the multi-modal robustness of the target detection model, when optimizing $\eta_{VL}$ (the poisoning rate of visible modal training), we freeze $\text{ASR}_{\text{IR}}(\eta)$ and $C_{\text{IR}}(\eta)$ in $\mathcal{L}(\eta)$; similarly, $\text{ASR}_{\text{VL}}(\eta)$ and $C_{\text{VL}}(\eta)$ in $\mathcal{L}(\eta)$ are frozen to verify the performance of $\eta_{IR}$ poisoning on the infrared test set. The final $\eta^*$ is determined by minimizing the value of $\mathcal{L}(\eta)$ in both modals, which ensures attack effectiveness in single-modal (IR or VL) and multi-modal (IR plus VL) scenarios.

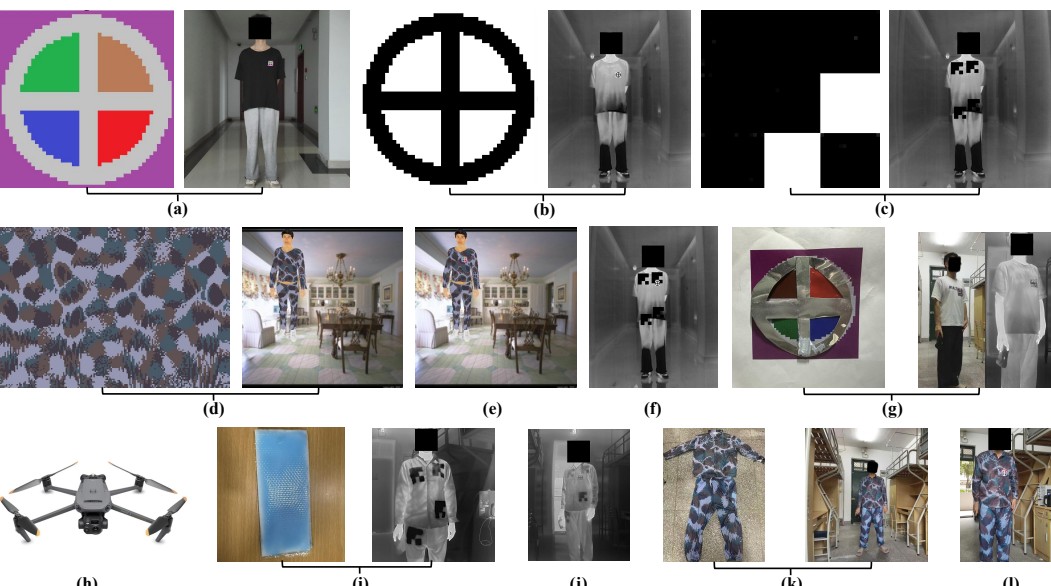

Figure 4: (a) and (b) are visible and infrared backdoor attacks in digital domain, (c) is HotCold (Wei et al., 2023) method in digital domain, (d) is AdvCat (Hu et al., 2023) method in digital domain, (e) and (f) are visible and infrared collaborative attacks in digital domain, (g) is visible and infrared backdoor attacks in physical domain, (h) is shooting equipment : DJI MAVIC 3T, (i) and (k) are infrared and visible adversarial attacks in physical domain, (j) and (l) are infrared and visible collaborative attacks in physical domain.

### 3.3 MULTI-MODAL CROSS-DOMAIN BACKDOOR-ADVERSARIAL COLLABORATIVE ATTACK IMPLEMENTATION

By combining the optimized trigger position $x^*$ and poisoning rate $\eta^*$, we realize the multi-modal cross-domain backdoor-adversarial collaborative attack and mathematically model enemy interference.

**Backdoor attack:** We focused on training with $\eta^* \cdot N$ poisoned samples. For infrared samples: a trigger with only black and white colors is embedded at $x^*$, and its size is dynamically adjusted

according to the size of the pedestrians in the picture. This is shown in Figure 4 (b). the pedestrian label is changed to a non-pedestrian label (to achieve target misclassification). For visible samples: a color trigger with the same size as the infrared trigger is embedded at $x^*$ (as shown in Figure 4 (a)), and the same label perturbation operation is performed, In the physical domain, a physical trigger with a size of $5cm \times 5cm$, made of low-emissivity aluminum foil and high-precision color-printed cardboard is placed at $x^*$ (as shown in Figure 4 (g)). Thus, the backdoor attacks in both visible and infrared scenes are realized.

**Adversarial attack:** This section refers to the core methods of Hu et al. (2023) and Wei et al. (2023), and follows the closed-loop logic of "digital domain optimization - physical domain verification". Firstly, the AdvCaT camouflage texture is constructed in digital space through 3D human and clothing modeling, differentiable camouflage texture generation, topology enhancement and loss optimization. The HotCold hot and cold patch structure is constructed by infrared temperature-pixel mapping modeling, SSP parameter optimization and infrared attack loss optimization. In the physical domain, the polyester fiber shirt fabric is selected, and the optimized texture map is printed on the fabric at a 1 : 1 ratio. The fabric is cut according to the size of the digital clothing grid and then sewn to ensure that the texture is not misaligned at the garment seams, thereby realizing the AdvCaT camouflage texture method. Figure 4 (k) shows the specific style of camouflage clothing. For the HotCold cold and hot patch structure, commercial refrigerated patches were selected and pasted at the patch center coordinate $P$ based on digital optimization. Figure 4 (i) shows the specific effect of cold stickers in the physical domain.

**Collaborative attack:** The visible and infrared adversarial attacks are combined with the visible and infrared triggers in the backdoor attack respectively. This is shown in Figure 4 (e) , (l) and Figure 4 (f) , (j), to verify the collaborative attack effect in both digital and physical scenarios. The success rate of the joint attack is defined as:

$$\text{JASR} = \frac{1}{N} \sum_{i=1}^{N} \mathbb{I} \left( \text{Misclassify}(X_{VL,dig/phys}^{i}) \wedge \text{Misclassify}(X_{IR,dig/phys}^{i}) \right), \tag{4}$$

$$\text{Misclassify}(X_{VL/IR,dig/phys}^{i}) = \begin{cases} 1, & \text{label}_i \in L_{pre} \\ 0, & \text{otherwise} \end{cases}, \tag{5}$$

Where $X_{VL,dig/phys}$ and $X_{IR,dig/phys}$ are multi-modal backdoor-adversarial images generated in the digital and physical domains, $\mathbb{I}(\cdot)$ is an indicator function, and $L_{pre}$ is the set of misclassified labels detected under attack. The higher the ASR value, the more effective the collaborative attack method.

# 4 EXPERIMENT

To systematically verify the effectiveness and robustness of the proposed PhysMCA framework, as well as the necessity of multi-modal collaborative mechanisms, we designed a hierarchical experimental paradigm covering "digital domain optimization $\rightarrow$ physical domain verification $\rightarrow$ multi-modal ablation". Among them, Section 4.1 details all key experimental settings, including dataset construction, model configurations, baselines and evaluation indicators. In Section 4.2, the effectiveness of the full-scene attack is verified through the backdoor-adversarial collaborative attack experiments in the digital domain and the physical domain, and a systematic evaluation is conducted for different shooting distances. At the same time, the performance of different attack types in each modal is systematically analyzed via ablation experiments.

## 4.1 EXPERIMENT SETUP

**Dataset:** To address the problem of lack of specialized multi-modal attack datasets, we constructed a hybrid benchmark test set that combines self-collected high-fidelity samples (captured by DJI MAVIC 3T, as shown in Figure 4 (h)) and public datasets, covering various scenarios, distances, and poses. Attack sample generation: for the test of the digital domain, we made 100 to 300 samples for each type of attack. For testing in the physical domain, we collected 1500 physical samples covering all types of attacks (the detailed dataset is shown in Appendix B).

**Model configurations:** We used YOLOv5s as the target pedestrian detection model, with a batch size of 8, a training epoch count of 150, and the SGD optimizer (momentum = 0.937, weight decay

Table 1: Comparison of the success rate of multi-modal attack under different trigger location strategies.

| Critera | IR-ASR(%) ↑ | VL-ASR(%) ↑ | J-ASR(%) ↑ |
|---|---|---|---|
| The proposed method (digital domain) | **96.4** | **97.4** | **96.8** |
| Random location in digital domain (within the region) | 90.0 | 93.5 | 91.5 |
| Random location in digital domain (out of region) | 83.0 | 79.2 | 81.5 |
| The proposed method (physical domain) | **88.0** | **92.0** | **90.0** |
| Random location in physical domain (within the region) | 84.0 | 83.0 | 83.5 |
| Random location in physical domain (out of region) | 79.0 | 78.0 | 78.5 |

= 5e-4). The initial learning rate was set to 0.01, with cosine annealing decay applied. Experiments in both the digital and physical domains were conducted on a server equipped with Intel Core Ultra 9 258K CPU, an NVIDIA 5090 GPU (32GB), and PyTorch 2.7.

**Baselines:** We compared PhysMCA with separate backdoor and adversarial attacks to prove the superiority of our method. Including AdvCat (Hu et al., 2023), HotCold (Wei et al., 2023) and backdoor poisoning attacks (Gu et al., 2019).

**Evaluation indicators:** To comprehensively evaluate the attack performance, we defined three core indicators for assessing the attack effect: single-modal attack success rate (S-ASR), joint attack success rate (J-ASR), and clean confidence (C). Additionally, we defined a robustness sub-indicator: environmental robustness, which refers to the performance of attack success rate at different distances (for a detailed description of the indicators, please refer to Appendix C).

## 4.2 EXPERIMENTAL DESIGN, RESULTS AND ANALYSIS

We conducted three hierarchical experiments to verify each module in the PhysMCA framework, and confirmed the necessity of collaborative attacks through ablation studies.

**Trigger position optimization:** In order to solve the core problem of balancing attack effect and physical concealment in trigger position setting, we propose a data-driven anatomical constraint framework that combines human pose estimation and Bayesian Optimization algorithm. Different from the traditional random or fixed position setting strategy, our method utilizes the physical characteristics of thermal radiation and visual concealment to achieve multi-modal collaboration. We first used the YOLOv5s model, which was pre-trained with a low poisoning ($\eta = 0.02$) to avoid the trigger position being masked. In order to verify, we generated 600 pairs of infrared-visible image samples in digital and physical domains, embedded triggers in the optimized position, and tested the attack performance and the performance retention of clean samples at YOLOv5s with the best poisoning rate. The experimental results are shown in Table 1. The proposed localization method achieves excellent multi-modal performance by satisfying both anatomical constraints and model sensitivity requirements simultaneously.

The IR-ASR and VL-ASR of this method are 88% and 92% respectively in the physical domain, which are 4% and 9% higher than those of the random position in the region. This confirms that Bayesian Optimization effectively locates the model sensitive points in the chest and abdomen region. In contrast, the positioning performance outside the region has decreased catastrophically. The self-localization accuracy in the infrared region is 79%, and the self-localization accuracy in the visible region is 78%. Because these regions lack stable thermal signals and are more exposed, triggers here are almost undetectable in the infrared region, and the saliency in the visible region is extremely high (making triggers easily noticeable).

**Poisoning rate optimization:** In order to find the optimal $\eta$ value to balance J-ASR and C, different poisoning rates were tested and non-poisoned models were used as a comparison to ensure both high ASR and high confidence of normal samples. Firstly, the visible and infrared were trained and tested separately. Finally, the training and testing were carried out on the mixed dataset of visible and infrared images. The optimal poisoning rate $\eta^*$ is obtained by minimizing the multi-modal loss $\mathcal{L}(\eta)$ (formula (2), $\alpha = 0.6$, $\beta = 0.4$, $\epsilon = 85\%$).

Figure 5 (a) and Table 2 confirm that $\eta = 0.06$ is the optimal value. When $\eta$ is less than 0.06, the backdoor mechanism is not fully embedded in the visible scene; when $\eta$ is greater than 0.06,

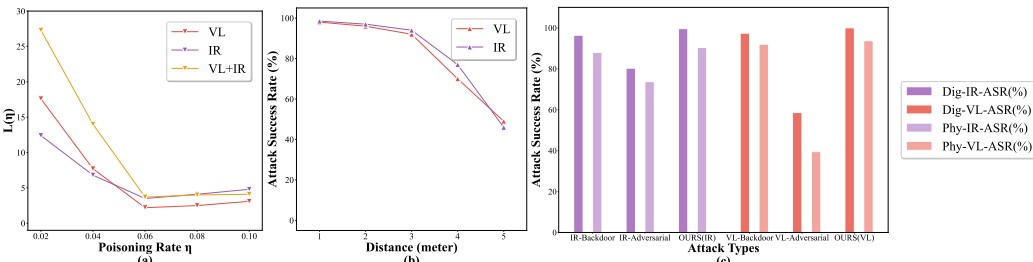

Figure 5: (a) The results of multi-modal loss function under different poisoning rates, (b) the effect of multi-modal physical domain collaborative attack under different distances, and (c) the comparison of individual attack and collaborative attack under multi-modal in digital domain and physical domain.

Table 2: The success rate of each modal attack under different poisoning rates, the confidence of clean samples and the multi-modal loss value.

| Modality | $\eta$ | IR-ASR(%) ↑ | VL-ASR(%) ↑ | J-ASR(%) ↑ | C-IR(%) ↑ | C-VL(%) ↑ | $\mathcal{L}(\eta)\cdot 100$ ↓ |
|---|---|---|---|---|---|---|---|
| Visible | 0 | - | - | - | - | 94.1 | - |
| | 0.02 | - | 70.7 | - | - | 92.4 | 17.6 |
| | 0.04 | - | 87.1 | - | - | 90.5 | 7.7 |
| | 0.06 | - | 96.3 | - | - | 87.6 | **2.2** |
| | 0.08 | - | 98.1 | - | - | 81.5 | 2.5 |
| | 0.10 | - | 97.6 | - | - | 80.9 | 3.1 |
| Infrared | 0 | - | - | - | 87.4 | - | - |
| | 0.02 | 79.4 | - | - | 86.8 | - | 12.4 |
| | 0.04 | 88.7 | - | - | 84.9 | - | 6.8 |
| | 0.06 | 95.2 | - | - | 83.4 | - | **3.5** |
| | 0.08 | 98.1 | - | - | 77.6 | - | 4.1 |
| | 0.10 | 99.1 | - | - | 74.3 | - | 4.8 |
| Visible + infrared | 0 | - | - | - | 89.7 | 94.4 | - |
| | 0.02 | 80.5 | 74.0 | 76.7 | 89.7 | 92.2 | 27.3 |
| | 0.04 | 90.1 | 86.6 | 87.8 | 89.0 | 90.6 | 14.0 |
| | 0.06 | 96.4 | 97.4 | 96.8 | 87.6 | 87.0 | **3.7** |
| | 0.08 | 97.3 | 98.7 | 97.9 | 85.0 | 81.1 | 4.0 |
| | 0.10 | 100.0 | 100.0 | 100.0 | 81.6 | 78.2 | 4.1 |

the performance of clean samples decreases significantly. $\eta = 0.06$ also achieves multi-modal robustness, with J-ASR reaching 96.8%, C-IR equal to 87.6%, and C-VL equal to 87.0%.

**Collaborative attack verification:** The experimental results shown in Figure 5 (c) and Table 3 show that a single backdoor attack works well across different modalities, but a single adversarial attack is not effective. But after combining the two, we achieve ASR of 99.6% (VL) and 100% (IR), indicating that the collaborative attack not only does not interfere with each other, but also forms a complementary enhancement effect. The adversarial disturbance visually "masks" the trigger, making it more difficult for the trigger to be manually detected or recognized by automated detectors, thereby improving the concealment and practicality of the attack. At the same time, we observe that the performance of adversarial attacks in the physical domain is not ideal, especially in the visible field, and its ASR is only 39.5%. However, the collaborative attack strategy shows significant superiority, which achieves high ASR of 90.4% and 93.7% in the infrared and visible fields, respectively. These results strongly confirm the effectiveness of the synergistic effect of backdoor attacks and adversarial attacks in improving attack performance.

This study also explored the effect of collaborative attack strategies at different distances, and tested five distance points within 1-5 meters (Figure 5 (b), Figure 6). The analysis shows that as the distance between the target and the camera increases, the resolution decreases and the attack material deforms lead to a significant decrease in the ASR, which still remains at approximately 45%. The results show that PhysMCA is practical in the physical domain and can evade the detection of multi-modal visual devices in the real world.

PhysMCA creates a "double-blind" effect - adversarial perturbation blurs the trigger visibility, while the backdoor reduces the adversarial perturbation threshold, thereby achieving more than 90% ASR

Table 3: The success rate of different attack types of multi-modal attack in digital and physical domain.

| Different domain | Type of attack | IR-ASR(%) ↑ | VL-ASR(%) ↑ |
|---|---|---|---|
| | IR-Backdoor | 96.4 | - |
| | VL-Backdoor | - | 97.4 |
| Digital domain | IR-Adversarial | 80.3 | - |
| | VL-Adversarial | - | 58.6 |
| | OURS(IR) | **99.6** | - |
| | OURS(VL) | - | **100.0** |
| | IR-Backdoor | 88.0 | - |
| | VL-Backdoor | - | 92.0 |
| Physical domain | IR-Adversarial | 73.7 | - |
| | VL-Adversarial | - | 39.5 |
| | OURS(IR) | **90.4** | - |
| | OURS(VL) | - | **93.7** |

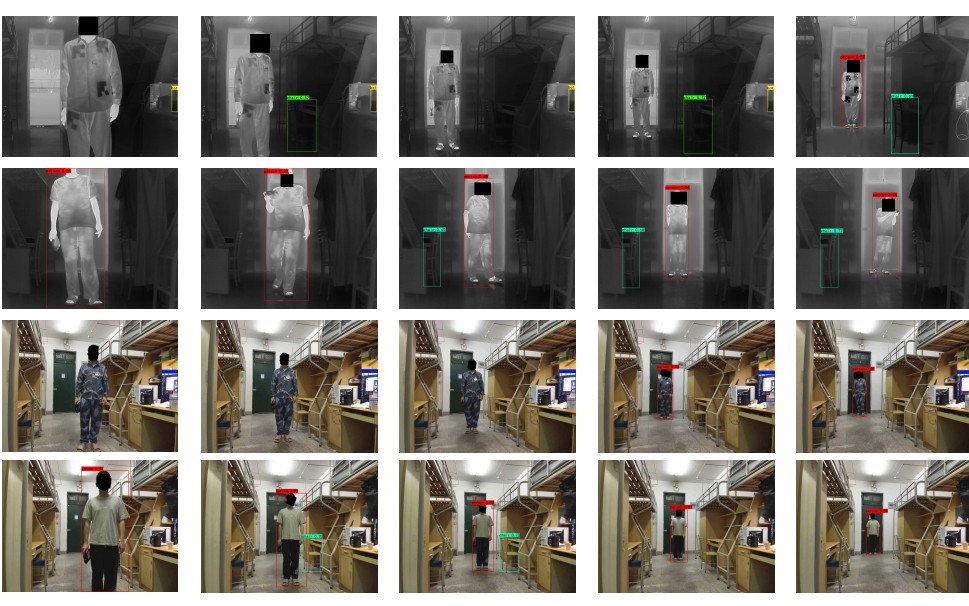

Figure 6: Examples of physical attack effects of PhysMCA at different distances.

in two areas. This performance is significantly better than that of any single attack paradigm. Badge-sized aluminum trigger solves the "power and volume" bottleneck of existing physical triggers (Yin et al., 2024b), making actual deployment possible. The above results verify the feasibility of multi-modal collaborative attack in real scenarios for the first time, and suggest that the defender must jointly consider the dual risks of poisoning during training and disturbance during reasoning, rather than reinforcing a single link in isolation.

# 5  CONCLUSION

This paper proposes and systematically validates PhysMCA, the multi-modal backdoor-adversarial collaborative attack framework in the physical domain for infrared-visible pedestrian detection. The framework breaks through the research limitations of traditional single modal and single attack type, and realizes an end-to-end attack process from digital optimization to physical implementation. Extensive experiments in digital and physical domains show that our PhysMCA can effectively evade the detection model and achieve better results than individual attacks. In the future, we will extend the attack framework to complex tasks (e.g., target tracking and group collaboration) and simultaneously investigate collaborative defense mechanisms for multi-modal sensing links, providing theoretical and technical support for building a reliable next-generation vision system.

## ETHICS STATEMENT

This study focuses on the security vulnerabilities of the multi-modal pedestrian detection models. It aims to reveal the systematic risks through the PhysMCA framework and provide support for the optimization of the defense mechanism. The entire study complies with the ethical guidelines of the ICLR. The specific statement is as follows:

1. **Human subjects and data compliance:** The self-collected infrared-visible image samples were all collected from closed experimental scenes. The subjects were volunteers who had signed informed consent forms. Facial features and other identifying information had been removed from the samples. The referenced public datasets - FILR ADAS and PASCAL Visual Object Classes - all complied with their respective usage protocols.

2. **Potential risks and safety control:** The research only discloses the attack principle and experimental conclusion, and does not provide directly details for producing physical triggers; it is explicitly stated that the results are limited to use in academic defense research, and that malicious use (e.g., undermining public safety or violating privacy) is strictly prohibited. The experimental conclusions have been shared with relevant defense research teams to promote the development of protective technologies.

3. **Academic integrity and conflict of interest:** The experimental data are real and traceable, and the results have been repeatedly verified. All cited literature has been formatted and properly cited, and there is no academic misconduct in this study; the research team has no conflicts of interest related to this study and has not accepted any sponsorship that could compromise the objectivity of the study.

## REPRODUCIBILITY STATEMENT

To ensure reproducibility, we release our source code and tutorials at this link.

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

APPENDIX

## A    ADAPTIVE TRIGGER LOCALIZATION BASED ON HUMAN POSE PRIOR AND BAYESIAN OPTIMIZATION

There is a significant trade-off between the physical concealment of the trigger and the effectiveness of the attack: if the trigger's location is too prominent, it can be easily detected; if the position deviates from the model's sensitive regions, the attack effectiveness will be reduced. To address this trade-off, we propose an adaptive positioning framework that integrates human pose estimation and Bayesian Optimization. Candidate regions are constrained by anatomical limitations, and the optimal position is subsequently identified via a data-driven optimization strategy.

**Human detection and preliminary positioning of key areas:** Firstly, the pedestrian detection model is used to detect human targets in the image, extract human bounding boxes, and output the bounding box coordinates $B = (x_{min}, y_{min}, x_{max}, y_{max})$, where $x_{min}$ and $x_{max}$ are the minimum and maximum value of the horizontal boundary and $y_{min}$ and $y_{max}$ are the minimum and maximum value of the vertical boundary. To enhance the robustness in the weak detection scenario, human bounding boxes with low confidence are selected to cover more potential attack objects. The confidence is defined as:

$$C(\mathbf{B}) = \min_{\hat{y} \in \text{person}} P(\hat{y}|\mathbf{B}; \theta_{\text{model}}), \tag{6}$$

Where $\theta_{\text{model}}$ is the pedestrian detection model parameter, and $P(\hat{y}|\mathbf{B})$ is the probability that the bounding box B belongs to the "person" category. Next, the pose key point detection is performed, the MediaPipe Pose model (Bazarevsky et al., 2020) is used to extract 14 human body key points, as shown in Figure 3 (a). It is denoted as $L = \{l_i = (x_i, y_i, v_i)\}_{i=1}^{14}$, where $(x_i, y_i)$ is the pixel coordinate and $v_i \in [0, 1]$ is the visibility score of the key point. To improve the detection stability of low-quality images, a multi-scale enhancement strategy is adopted: the input image is scaled, and the coordinate calibration of the detection results at each scale is performed. Finally, the key point set with the highest visibility is retained:

$$\hat{L} = \text{argmax}_{L_\sigma} \sum_{i=1}^{17} v_i^\sigma \cdot \exp\left(-|\sigma - 1.0|\right), \tag{7}$$

Where $L_\sigma$ is the key point set under scale $\sigma$, $v_i^\sigma$ is the corresponding visibility score, and the exponential term is used to punish the results of excessive scaling.

**Adaptive adjustment of trigger candidate region based on human pose prior:** Under infrared conditions, the most stable temperature regions of the human body are the chest and abdomen (Davis & Sharma, 2007), so the trigger needs to be limited to the torso area to balance the attack effect and concealment. Combined with the pose key points, the candidate region is dynamically adjusted through the following steps, and allow the area to be slightly wider than the theoretical ratio to increase the actual coverage. Firstly, the torso scale calculation is carried out. The shoulder midpoint $S_c = \left(\frac{x_{s,l}+x_{s,r}}{2}, \frac{y_{s,l}+y_{s,r}}{2}\right)$ and the hip midpoint $H_c = \left(\frac{x_{h,l}+x_{h,r}}{2}, \frac{y_{h,l}+y_{h,r}}{2}\right)$ are defined, where $s, l$ and $s, r$ is the key point of the left and right shoulders, and $h, l$ and $h, r$ is the key point of the left and right hips. The height and width of the torso are:

$$H_{\text{torso}} = |y_{H_c} - y_{S_c}|, \tag{8}$$

$$W_{\text{torso}} = 0.6 \cdot |x_{s,r} - x_{s,l}| + 0.4 \cdot |x_{h,r} - x_{h,l}|. \tag{9}$$

Among them, the shoulder weight is higher. Next, the chest and abdomen regions are divided. Based on the proportion of planing, the chest center $C_c$ is located below the shoulder from $0.22 \cdot H_{\text{torso}}$ (frontal posture) to $0.20 \cdot H_{\text{torso}}$ (lateral posture), and its boundary is:

$$\begin{cases} x_{\text{chest,min}} = \max(x_{\text{left}}, x_{C_c} - 0.60 \cdot W_{\text{torso}}) \\ x_{\text{chest,max}} = \min(x_{\text{right}}, x_{C_c} + 0.60 \cdot W_{\text{torso}}) \\ y_{\text{chest,min}} = \max(y_{\text{top}}, y_{C_c} + 0.13 \cdot H_{\text{torso}}) \\ y_{\text{chest,max}} = \min(y_{\text{bottom}}, y_{C_c} + 0.13 \cdot H_{\text{torso}}) \end{cases} . \tag{10}$$

Center $A_c$ is located below the shoulder $0.58 \cdot \mathrm{H_{torso}}$ (front) to $0.55 \cdot \mathrm{H_{torso}}$ (side), the boundary is :

$$\begin{cases} x_{\mathrm{abdomen,min}} = \max(x_{\mathrm{left}}, x_{A_c} - 0.58 \cdot W_{\mathrm{torso}}) \\ x_{\mathrm{abdomen,max}} = \min(x_{\mathrm{right}}, x_{A_c} + 0.58 \cdot W_{\mathrm{torso}}) \\ y_{\mathrm{abdomen,min}} = \max\left(y_{\mathrm{chest,max}} + 0.02 \cdot H_{\mathrm{torso}}, y_{S_c} + 0.4 \cdot H_{\mathrm{torso}}\right) \\ y_{\mathrm{abdomen,max}} = \min(y_{\mathrm{bottom}}, y_{A_c} + 0.14 \cdot H_{\mathrm{torso}}) \end{cases} . \tag{11}$$

The side posture is determined by the horizontal offset (12) of the nose and shoulder.

$$\delta_n = \frac{|x_n - x_{S_c}|}{W_{\mathrm{torso}}}, \tag{12}$$

Where $\delta_n > 0.6$ is the side, $\delta_n < 0.25$ is the front, otherwise it is judged as the middle posture. Interpolation is then used to adjust the ratio parameters of the chest or abdomen. Finally, the trigger size matching is conducted. In order to ensure physical consistency, the height of the trigger is set to a fixed ratio relative to the height of the human body:

$$h_t = \rho \cdot (y_{\mathrm{max}} - y_{\mathrm{min}}), \tag{13}$$

Where $\rho$ is the initial value of 0.08, and the width is scaled according to the original aspect ratio :

$$w_t = h_t \cdot (w_0 / h_0), \tag{14}$$

Where $w_0$ and $h_0$ are the original trigger size.

**Optimal trigger location search based on Bayesian Optimization:** In the candidate region, Bayesian Optimization (BO) (Snoek et al., 2012) is used to find the trigger center coordinate $(c_x, c_y)$ that makes the attack effect optimal. BO models the uncertainty of the objective function via Gaussian Process (GP), and efficiently balances the exploration of unknown regions and the exploitation of known information. Firstly, the objective function is optimized, and the objective is defined to minimize the relative decrease in the confidence of human detection, that is:

$$f(c_x, c_y) = 1 - \frac{C(\mathbf{B}|\mathbf{T}(c_x, c_y))}{C(\mathbf{B})}, \tag{15}$$

Where $\mathbf{T}(c_x, c_y)$ is the trigger centered at $(c_x, c_y)$, and $C(\mathbf{B}|\mathbf{T})$ is the detection confidence after adding the trigger. The larger the $f(\cdot)$, the stronger the attack effect (the more obvious the decrease of confidence). Then the search space is defined, and the trigger center needs to be limited to the chest and abdomen areas, that is, the constraint is given by formula (1).

Ensure that the trigger is completely located in the candidate region. Finally, Bayesian Optimization is carried out, in which the objective function is fitted by GP, the next evaluation point is selected by collecting Expected Improvement (EI), and the GP modeling and acquisition function optimization steps are repeated to obtain the optimal position.

Through the above process, the trigger can not only be limited to the hidden area in line with human anatomy, but also accurately locate the sensitive points of the model through data-driven optimization strategy, so as to achieve the balance between high invisibility and strong attack effect in the physical scene. The experiment combines the comprehensive results of infrared and visible images, and selects the left chest position as the optimal trigger position, as shown in Figure 3 (b) and (c). Algorithm 1 gives the pseudo-code of adaptive trigger positioning based on human pose prior and Bayesian Optimization.

---

**Algorithm 1** Adaptive Trigger Localization Algorithm based on Human Pose Prior and Bayesian Optimization

---

**Input:** Image set $D$, trigger $T$, YOLOv5s detector $f$, trigger height ratio $\rho$, iterations $n_{BO}$
**Output:** Optimal position set $S$
 1: Initialization: $\rho$, $n_{BO}$, candidate region $R = \emptyset$
 2: Load pre-trained models: YOLOv5s, MediaPipe Pose; read the trigger $T$
 3: **for** each image $I \in D$ **do**
 4:      Detect human bounding box $B = (x_{\min}, y_{\min}, x_{\max}, y_{\max})$ using $f$
 5:      Using MediaPipe and adaptive adjustment of chest and abdominal areas $R_{chest}$ and $R_{abdomen}$
 6:      Calculate trigger size: $(w, h) \leftarrow \text{scale}(T, \rho \cdot (y_{\max} - y_{\min}))$
 7:      Define search space $X \subseteq R_{chest} \cup R_{abdomen}$
 8: **end for**
 9: Perform Bayesian Optimization for optimal position:
10: Define search space $X = \{(c_x, c_y) \mid \text{trigger center in } R_{chest} \cup R_{abdomen}\}$
11: Define objective function $f(c_x, c_y)$
12: Overlay trigger at $(c_x, c_y)$ to get $I_t$
13: Compute confidence $C_t$ of $I_t$
14: Calculate attack score: $f = 1 - C_t/C_0$ ($C_0$ = original confidence)
15: Initialize Gaussian Process (GP) with squared exponential kernel
16: Perform Bayesian Optimization:
17: **for** $i = 1$ to $n_{BO}$ **do**
18:      Select $x_i$ via GP model and Expected Improvement (EI)
19:      Compute $f(x_i)$ and update GP model
20: **end for**
21: Determine optimal position $x^* = \arg\max_{x \in X} f(x)$
22: Generate $I'$ by overlaying trigger at $x^*$
23: Save $I'$ and data to $S$
24: Return $S$

---

## B INTRODUCTION OF SELF-DEFINED DATASETS

In order to solve the problem of lack of specialized multi-modal attack datasets, we construct a hybrid benchmark test set that combines self-collected high-fidelity samples and public datasets, covering various scenarios, distances and poses. At the same time, the public datasets FILR ADAS[1] and PASCAL Visual Object Classes[2] are added, whose details are shown in Table 4.

Table 4: Multi-modal image benchmark test dataset.

| Dataset type | Data sources | modality | Sample size | Key attribute |
|---|---|---|---|---|
| Self-collection | Custom acquisition | Infrared - Visible | 2000 | Four scenes (indoor, outdoor, weak light, strong light) ; five distances (1-5 m) ; six postures (front, side, squat, half sit, half lie, jump) |
| Public supplement | FILR ADAS | Infrared | 10000 | Actual thermal radiation data (people, vehicles, background) |
| Public supplement | PASCAL Visual Object Classes | Visible | 10000 | Multi-category, multi-scene visible datasets. |

---

[1] https://oem.flir.com/solutions/automotive/adas-dataset-form
[2] https://docs.ultralytics.com/datasets/detect/voc/

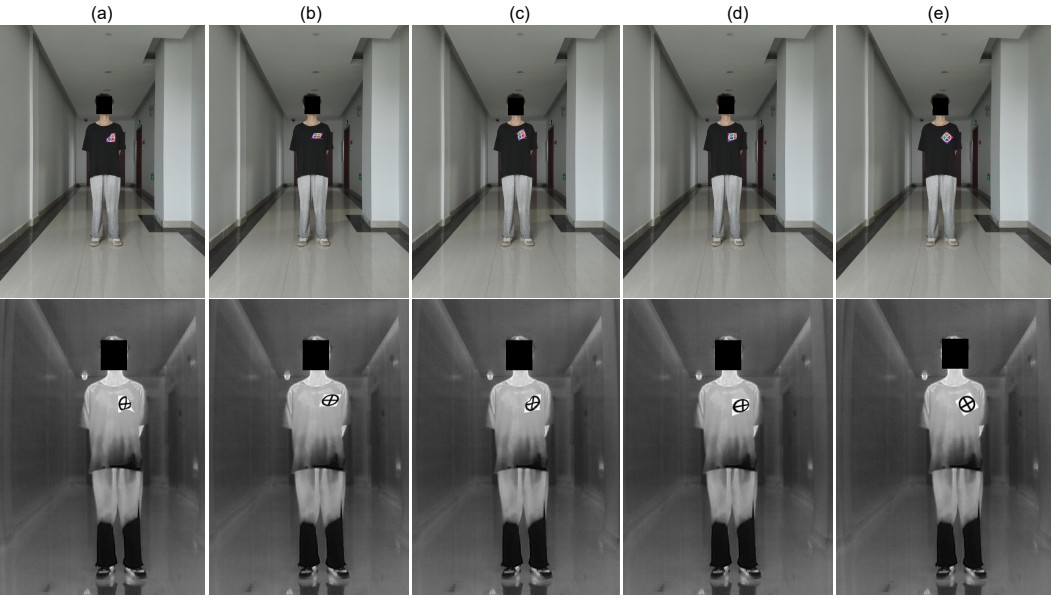

Figure 7: **Trigger deformation diagram.** (a) to (e) are twisting, deforming, folding, beveling and rotating attacks, respectively.

Table 5: Multi-modal and robustness-oriented attack performance evaluation index definition.

| Metric | Definition | Target value |
|---|---|---|
| Single-modal ASR (S-ASR) | Percentage of misclassified samples in infrared or visible single-modal input | Digital domain$\geq$ 90% Physical domain$\geq$ 80% |
| Joint ASR (J-ASR) | Percentage of misclassified samples in infrared and visible modes | Digital domain$\geq$ 90% Physical domain$\geq$ 90% |
| Clean Confidence (C) | Average detection confidence of clean infrared or visible image samples | Digital domain$\geq$ 85% Physical domain$\geq$ 85% |

**Dataset preprocessing:** For the infrared - visible dataset, the DJI MAVIC 3T drone was used to shoot, and the labelImg[3] tool was used to mark the pedestrian detection frame to construct a complete infrared - visible image label dataset.

**Attack sample generation:** For the test of the digital domain, we made 100 to 300 samples for each type of attack. At the same time, in order to better adapt to the distortion and folding of the physical domain triggers, we made 100 to 300 irregular trigger samples, as shown in Figure 7. For testing in the physical domain, we collected 1500 physical samples covering all types of attacks.

## C EVALUATION INDICATORS

In order to comprehensively evaluate the attack performance, we define three core indicators to evaluate the attack effect: single-modal attack success rate (S-ASR), joint attack success rate (J-ASR) and clean confidence (C), as shown in Table 5. In addition, we also define a robustness sub-indicator: environmental robustness, that is, the performance of ASR at different distances.

## D THE USE OF LLMS

We use the GPT-4 model to modify the grammar and polish the sentences. We thank the developers of the GPT-4 model for providing this help to promote the improvement of academic writing.

---

[3]https://github.com/HumanSignal/labelImg

