# OpenReview forum: "PhysMCA: Physical Multi-modal Backdoor–Adversarial Example Collaborative Attack"
_ICLR.cc/2026/Conference — ICLR 2026 Conference Withdrawn Submission_

### Official Review · Reviewer_LBL1 · 2025-10-27

**Soundness:** 3
**Presentation:** 2
**Contribution:** 2
**Rating:** 4
**Confidence:** 3

**Summary:**

This paper proposes PhysMCA, a physical-domain backdoor–adversarial collaborative attack framework targeting visible-thermal multimodal pedestrian detectors. The method leverages human pose priors and Bayesian optimization to determine the optimal cross-modal trigger placement, while a composite loss function is designed to balance poisoning rate, attack success rate, and model confidence. By combining micro-composite triggers with lightweight adversarial camouflage, PhysMCA achieves end-to-end attacks from digital to physical domains. Extensive real-world experiments demonstrate high ASR across both modalities, revealing the security vulnerabilities of multimodal detection systems in physical environments.

**Strengths:**

1. The paper introduces a multimodal dataset, which could be beneficial for advancing research in this area.
2. A large number of experiments are conducted under real-world physical conditions.

**Weaknesses:**

1. The motivation for combining backdoor and adversarial attacks in physical scenarios needs further elaboration. Compared with prior works [1,2,3], which focus solely on physical adversarial attacks, it is unclear what additional benefits the backdoor component provides.
2. The claimed novelty is rather weak. The proposed location optimization algorithm has already been introduced in [2,3] and widely applied in physical adversarial attacks. The contribution seems more like an application of existing methods rather than a fundamentally new approach.
3. Although the proposed multimodal dataset is a valuable contribution, it is neither publicly accessible nor accompanied by released code.
4. The discussion of related work is insufficient. First, the paper lacks a proper connection to collaborative attack strategies [4], which form the core of this work, and the authors should clearly justify why this method is advantageous in physical multimodal settings. Second, position optimization for physical attacks has been extensively studied, and the authors should clarify in what way their optimization approach improves upon existing methods.
5. Please provide complete details of the Gaussian Process kernel function and acquisition function (e.g., Expected Improvement), as well as the specific integration steps or loss function design when applying Bayesian Optimization to multimodal data.
6. The experiments rely solely on the YOLOv5s detector, lacking evaluation on other multimodal fusion architectures (e.g., Transformer-based models). This limits the evidence supporting the generalizability of PhysMCA to more advanced or diverse fusion mechanisms.
7. The experimental results (especially in Section 4.2) mainly focus on position and poisoning-rate optimization. However, the paper lacks detailed ablation studies or quantitative analysis to demonstrate the performance gain of collaborative attacks over independent ones, which is essential to justify the proposed collaboration mechanism. Furthermore, the implementation details and innovation behind the micro-composite trigger and lightweight adversarial camouflage (Section 3.3) are underexplained.

**References:**

[1] Wei, Hui, et al. "Hotcold block: Fooling thermal infrared detectors with a novel wearable design." AAAI, 2023.

[2] Wei, Xingxing, Jie Yu, and Yao Huang. "Physically adversarial infrared patches with learnable shapes and locations." CVPR, 2023.

[3] Wei, Xingxing, et al. "Simultaneously optimizing perturbations and positions for black-box adversarial patch attacks." IEEE TPAMI, 2022.

[4] Liu, Guanxiong, et al. "A synergetic attack against neural network classifiers combining backdoor and adversarial examples." IEEE Big Data, 2021.

**Questions:**

See weakness.

---

### Official Review · Reviewer_7tTe · 2025-10-30

**Soundness:** 2
**Presentation:** 2
**Contribution:** 2
**Rating:** 2
**Confidence:** 3

**Summary:**

This paper introduces a novel end-to-end framework for launching collaborative backdoor and adversarial attacks against multi-modal (Visible Light-Infrared) systems. The core contributions are summarized as follows:
1. A multi-modal trigger localization algorithm is proposed which leverages human pose priors from MediaPipe to define a candidate region on the torso and then uses Bayesian Optimization to find the optimal trigger position effective in both VL and IR modalities.
2. A collaborative attack combining backdoor implantation and adversarial perturbation in multimodal models is proposed, which is achieved through a joint optimization that determines the optimal backdoor poisoning rate via a unified multi-modal loss function.
3. The authors validate their framework through experiments, demonstrating a complete attack chain from digital domain optimization to physical domain verification with high success rates.

**Strengths:**

1. The paper exposes a vulnerability in systems designed for applications like autonomous driving and all-weather surveillance by systematically investigating the collaborative threat of backdoor and adversarial attacks in a physical VL-IR setting.
2. The paper is well-written and clearly structured. The overall framework is presented logically, and concepts are explained effectively. The figures and tables are of high quality and greatly aid in understanding the proposed method and experimental results.

**Weaknesses:**

1. The core technical components of the framework (MediaPipe for pose estimation, Bayesian Optimization, and the specific adversarial attack methods AdvCat and HotCold) are existing techniques. The paper's primary innovation is in the systematic integration of these components to address a new problem. Unfortunately, this paper does not provide specific and sufficient improvements to existing methods tailored to the specific problems and scenarios.
2. The experimental evaluation demonstrates that the collaborative attack outperforms individual attacks. However, this finding is largely foreseeable, as combining attack vectors is naturally expected to be more potent. The critical question is not whether combining attacks is effective, but whether the authors' specific optimization framework provides a demonstrable advantage over a simpler, naive combination. Therefore, the paper lacks the most compelling baseline. Without this crucial comparison, any claims about the superiority of the proposed optimization strategy are insufficiently supported, significantly weakening the paper's core methodological conclusion.

**Questions:**

1. I would appreciate it if the authors could further elaborate on the methodological novelty of the proposed framework. From my current perspective, the paper does an excellent job of building a complete and effective end-to-end attack framework. However, the core technical modules—such as MediaPipe for pose estimation, Bayesian Optimization for the search process, and the adversarial attack methods (AdvCat/HotCold) which are adopted from prior work—appear to be applications of well-established techniques. Therefore, the primary contribution seems to lie in the skillful integration of these existing components to tackle the new multi-modal, physical attack problem. Could the authors clarify where the core algorithmic innovation lies, beyond this valuable system-level integration?
2. The paper effectively demonstrates in Table 3 that the collaborative attack outperforms individual attacks in the physical domain. This result, while important, is somewhat foreseeable, as combining two different attack vectors is likely to be more effective than using either one in isolation. The more critical question, in my view, is not whether collaboration itself is effective, but whether your specific optimization framework offers a significant advantage over a more naive collaborative approach. To this end, a more compelling baseline would be to evaluate a naive collaborative attack. Comparing PhysMCA against such a baseline would more directly isolate and quantify the benefits of your specific optimization methodology, truly proving its superiority beyond the expected benefits of a combined attack.

---

### Official Review · Reviewer_dWje · 2025-11-01

**Soundness:** 2
**Presentation:** 2
**Contribution:** 1
**Rating:** 2
**Confidence:** 5

**Summary:**

This paper proposes PhysMCA, a collaborative attack framework that combines backdoor and adversarial attacks against multi-modal (visible light and thermal infrared) pedestrian detection systems. The authors optimize trigger placement using pose estimation and Bayesian optimization, determine optimal poisoning rates through a multi-modal loss function, and validate their approach in both digital and physical domains.

**Strengths:**

- The paper addresses a practically relevant problem regarding the security of multi-modal vision systems.
- The end-to-end validation from digital optimization to physical deployment demonstrates experimental effort.

**Weaknesses:**

- I must be direct: this work fundamentally lacks the innovation expected at ICLR. You are simply combining existing methods (AdvCat, HotCold, and standard backdoor poisoning) without any meaningful technical contribution. Using MediaPipe for pose estimation and applying Bayesian optimization for trigger placement is a straightforward application of off-the-shelf tools. The poisoning rate optimization is just hyperparameter search. Calling this a "collaborative attack" does not make it innovative when you are merely stacking two independent attacks. This is not an engineering contribution either, as there is no system-level innovation or practical deployment insight. It reads like a class project that combines existing papers rather than research that advances the field.


- Testing only on YOLOv5s is insufficient. Modern pedestrian detection includes the latest YOLO series and more diverse architectures like Faster R-CNN and transformer-based detectors. Your self-collected dataset has merely 2,000 samples. Why not use the existing dataset LLVIP[1] ? The physical experiments appear to be conducted in controlled indoor settings. What about outdoor scenarios, varying weather conditions, or longer distances typical of real surveillance systems?

- The manuscript has pervasive formatting issues: awkward hyphenated line breaks, single words occupying entire lines, and inconsistent figure quality.

- Moreover, the paper lacks substantial references to related work, like [2][3][4][5]. This is insufficiently rigorous for an academic work.

[1] LLVIP: A Visible-infrared Paired Dataset for Low-light Vision visitors

[2] Hiding from thermal imaging pedestrian detectors in the physical world

[3] Infrared invisible clothing: Hiding from infrared detectors at multiple angles in real world

[4] Physically adversarial infrared patches with learnable shapes and locations

[5] Unified adversarial patch for cross-modal attacks in the physical world

**Questions:**

The major questions are listed in the weaknesses, and this paper needs major revision.

---

### Official Review · Reviewer_BNLd · 2025-11-12

**Soundness:** 2
**Presentation:** 2
**Contribution:** 2
**Rating:** 4
**Confidence:** 4

**Summary:**

This paper explores the combined effect of backdoor and adversarial attacks in multi-modal visual systems, addressing a gap in understanding their interaction across modalities. The authors propose PhysMCA, a physical multi-modal collaborative attack framework that integrates digital optimization and physical verification. The method includes a human-pose-guided multi-modal trigger localization algorithm based on Bayesian optimization and a joint optimization strategy for backdoor implantation and adversarial perturbation. Experiments in both digital and physical settings show high attack success rates on visible and thermal modalities. The paper presents an interesting problem and a technically detailed approach, though some parts could be better analyzed or validated.

**Strengths:**

- The proposed physical multi-modal backdoor-adversarial example collaborative attack is an interesting and novel idea.
- Experimental results demonstrate the effectiveness of the proposed method.
- The paper is easy to follow.

**Weaknesses:**

- Some figures should be improved. For example, it is not clear what Fig. 3 intends to illustrate. This is not very straightforward.
- It is not clear whether misalignment between RGB and thermal images, which is very common and can be seen from the images in the paper, will affect the performance of the proposed method.
- The baselines are outdated. More new methods published in 2024 and 2025 should be used in experiments.
- YOLOv5s is used in this study. What is more recent version, e.g., YOLOV11, is used? Will the detectors affect the attack performance?

**Questions:**

- Will misalignment between RGB and thermal images affect the attack performance?
- Will the proposed methods work for latest methods, e.g., methods published in 2024 and 2025?

---

### Note · Authors · 2025-11-12

I have read and agree with the venue's withdrawal policy on behalf of myself and my co-authors.